# Global implementation survey of Integrated Management of Childhood Illness (IMCI): 20 years on

Cynthia Boschi-Pinto,[1,2] Guilhem Labadie,[1] Thandassery Ramachandran Dilip,[1] Nicholas Oliphant,[3,4] Sarah L Dalglish,[1] Samira Aboubaker,[1] Olga Adjoa Agbodjan-Prince,[5] Teshome Desta,[6] Phanuel Habimana,[7] Betzabe Butron-Riveros,[8] Jamela Al-Raiby,[9] Khalid Siddeeg,[9] Aigul Kuttumuratova,[10] Martin Weber,[10] Rajesh Mehta,[11] Neena Raina,[11] Bernadette Daelmans,[1] Theresa Diaz[1]

For numbered affiliations see end of article.

**Correspondence to**
Dr Cynthia Boschi-Pinto;
cboschi200@gmail.com

## ABSTRACT

**Objective** To assess the extent to which Integrated Management of Childhood Illness (IMCI) has been adopted and scaled up in countries.

**Setting** The 95 countries that participated in the survey are home to 82% of the global under-five population and account for 95% of the 5.9 million deaths that occurred among children less than 5 years of age in 2015; 93 of them are low-income and middle-income countries (LMICs).

**Methods** We conducted a cross-sectional self-administered survey. Questionnaires and data analysis focused on (1) giving a general overview of current organisation and financing of IMCI at country level, (2) describing implementation of IMCI's three original components and (3) reporting on innovations, barriers and opportunities for expanding access to care for children. A single data file was created using all information collected. Analysis was performed using STATA V.11.

**Participants** In-country teams consisting of representatives of the ministry of health and country offices of WHO and Unicef.

**Results** Eighty-one per cent of countries reported that IMCI implementation encompassed all three components. Almost half (46%; 44 countries) reported implementation in 90% or more districts as well as all three components in place (full implementation). These full-implementer countries were 3.6 (95% CI 1.5 to 8.9) times more likely to achieve Millennium Development Goal 4 than other (not full implementer) countries. Despite these high reported implementation rates, the strategy is not reaching the children who need it most, as implementation is lowest in high mortality countries (39%; 7/18).

**Conclusion** This survey provides a unique opportunity to better understand how implementation of IMCI has evolved in the 20 years since its inception. Results can be used to assist in formulating strategies, policies and activities to support improvements in the health and survival of children and to help achieve the health-related, post-2015 Sustainable Development Goals.

## INTRODUCTION

WHO and Unicef introduced Integrated Management of Childhood Illness (IMCI) in the mid-1990s as a strategy to improve child survival in countries with more than 40 deaths per 1000 live births and provide integrated prevention, treatment and care for the sick child.[1] The strategy was later expanded to include care for sick newborns under 1 week of age, and has been regularly updated to reflect advancements in technical knowledge. Over 100 countries have adopted IMCI and implemented, in whole or in part, its three components: (1) improving health worker skills, (2) strengthening health systems and (3) improving family and community practices. Evidence suggests that IMCI has contributed to reductions in child mortality over the era of the Millennium Development Goals (MDGs), and a recent Cochrane review found the strategy was associated with a 15% reduction in child mortality when activities were implemented in health facilities and communities.[2]

Initial reviews of IMCI implementation, notably the 2003–2005 Multi-Country

### Strengths and limitations of this study

► This survey provides a unique opportunity to better understand how implementation of Integrated Management of Childhood Illness has evolved in the 20 years since its inception and help steer the direction of future child health strategies.

► Limitations arising from study design include the fact that respondents may not feel comfortable providing answers that present the country in an unfavourable manner and therefore may not provide accurate answers.

► The WHO African region is over-represented in the survey responses, but disaggregation by mortality level and income groups reduces the effect of this over-representation on interpretation of results.

Evaluation[3] and the 2003 Analytic Review,[4] used in-depth looks at specific country contexts to draw broader lessons about implementation dynamics. WHO has also regularly surveyed countries about their adoption and implementation of the IMCI strategy, and has occasionally published maps showing the extent of implementation. However, it has not provided a comprehensive description of global implementation worldwide, with details on country variations, implementation barriers and facilitators, local innovations and other particulars. Meanwhile, Unicef has run a number of surveys on countries' adoption and implementation of integrated community case management (iCCM), a strategy similar to IMCI focusing on provision of care for sick children by community health workers (CHWs), however the data produced focused mainly on implementation in sub-Saharan African countries.[5]

As part of the Strategic Review of IMCI,[6 7] a global survey was carried out in April–June 2016 to assess the extent to which this strategy has been adopted and scaled up in countries over the past two decades. This article reports on its main results and provides an overview of the current status of global implementation of IMCI as informed by countries. The article also presents the main strengths and barriers in IMCI implementation, as stated by respondents, and suggested ways forward.

## METHODS
### Study design
This was a cross-sectional self-administered global survey carried out from April to June 2016.

### Survey instrument
The survey instrument (online supplementary annex 1) was developed by WHO with inputs from Unicef and was tested in three different WHO regions to ensure questions were clear and appropriate, and to estimate the time needed to complete the questionnaire. The final version was made available in all six UN languages (Arabic, Chinese, English, French, Spanish and Russian) and included mostly closed-ended questions. Countries identified the principal strengths of IMCI and barriers to its implementation by agreeing or disagreeing with a suggested list, and answered a few open-ended questions about future actions to sustain and improve implementation.

### Variables
Questions and data analysis focused on variables providing: (1) a general overview of the current level of organisation and financing of IMCI in countries; (2) a description of implementation of IMCI's three original components and of additional ones, such as iCCM; and (3) the main innovations, barriers and opportunities in expanding access to care for the sick child.

### Study size and participants
Questionnaires were sent to a total of 130 WHO member states by the six WHO Regional Offices (ROs) which then implemented the survey through WHO country offices (COs) and provided support and follow-up. ROs had different approaches when sending out questionnaires to countries, reflecting specificities of each region. In the African region, questionnaires were sent to all 47 WHO member states. Four countries (Algeria, Cape Verde, Mauritius and Seychelles) responded that they had not adopted IMCI and therefore did not participate in the survey. Responsible officers from South Sudan and São Tomé & Principe did not respond to the survey. In the WHO Region of the Americas, questionnaires were sent to all 33 low-income and middle-income countries (LMICs) in the region, of which 19 (58%) responded. Four countries (Antigua & Barbuda, St Lucia, St Vincent & Grenadines and Venezuela) responded that they had not adopted IMCI; therefore 15 countries were included in the survey. Questionnaires were sent to 16 WHO countries in the Eastern Mediterranean region known to have adopted and implemented IMCI. Responses were received from 14 countries (88%). Of the 53 countries in the WHO European region, 15 have introduced and/or implemented IMCI at different points in time and on varying scale. Survey questionnaires were sent to seven WHO member states where WHO National Professional Officers were present, and six responded (86%). In the South-East Asia region, questionnaires were sent to all 11 WHO member states, of which nine (82%) responded. Finally, questionnaires were sent to 16 LMICs in the Western Pacific region, and 10 (69%) were included in the final detailed analysis. Responses were provided by in-country teams of IMCI experts with country experience, including varying combinations of representatives of the ministry of health (MoH) and WHO and Unicef COs, or in some instances just representatives of the MoH. Responses were tracked by WHO ROs to maximise return and clarify inconsistencies or omissions. Completed questionnaires were then returned from countries to WHO Headquarters.

### Data source and analysis
Data from the Excel questionnaires were initially converted into an Excel database. Responses were checked for skip patterns and missing data were recoded. Clarification was sought from countries with missing data, of which some responded. Analyses included descriptive statistics to synthesise and organise information using numerical procedures (simple frequency and means in univariate and bivariate tables), and/or graphic techniques to describe the characteristics of IMCI in the respondent countries. Data analysis was disaggregated by levels of under-five mortality rates (U5MRs), gross national income (GNI) per capita and extent of IMCI implementation. Countries were categorised into low ($\leq$40 deaths per 1000 live births), middle (40–80 deaths per 1000 live births) and high mortality (80+ deaths per 1000 live births). Countries were also classified by income level (low, middle and high) according to current GNI per capita (2016) using World Bank data,[8] and as high

(>90% of districts reported to have implemented IMCI), middle (50%–90% of districts reported to have implemented IMCI) and low implementers (<50% of districts reported to have implemented IMCI). Information on under-five mortality (numbers and rates) for 2015 was obtained from the United Nations Inter-agency Group for Child Mortality Estimation report on levels and trends in child mortality.[9] Information on under-five population for the year 2015 was obtained from the UN Population Division.[10] A single data file was created containing all information collected. Responses to precoded questions are presented in this paper. Analysis was performed using STATA V.11.0 software. More detailed methods are described elsewhere.[11]

### Patient involvement

This is a self-administered key-informant survey carried out at national level and no individual patient data were included.

### RESULTS

#### Participants and descriptive data

Questionnaires were sent to 130 WHO member states with a response rate of 80% (online supplementary table 1). Of the 104 countries that responded to the survey, eight reported no IMCI in the country, leaving 96 countries for the general analysis. One country was dropped from the analysis because IMCI was not reported to be part of the national child health plan or strategy, no district was reported to be implementing IMCI and none of the three components was reported to be in place (online flow diagram). Of the 95 countries retained for final analysis

(figure 1), only two (Oman and Uruguay) are classified as high-income countries according to the World Bank; 64 are classified as middle-income and 29 as low-income. These 95 countries are home to 82% of the global under-five population and account for 95% of the 5.9 million deaths that occurred among children less than 5 years of age in 2015.

The instruction that questionnaires be completed by a three-person in-country team (from MoH, WHO and Unicef) was followed in only five countries. In the majority of countries only the MoH provided answers (online supplementary table 2). For three countries, information on respondents was not available.

Eighty-nine per cent (82 countries) of the 94 responding countries had an IMCI focal point at national level, whereas only 68% (62/91) had one at the subnational (regional or district) level. Among high mortality countries, 88% (17/20) had a national focal point but less than half this proportion (42%; 8/19) had subnational focal points. The presence of regional/district focal points was much higher in countries with U5MR below 80 per 1000 live births (75%; 54/72) than in countries with higher U5MR. The government is generally reported to be the principal funding source for IMCI expenditures at first-level health facilities and in the community, except for health worker training and per diems, for which donors constituted the main sources of funding. In figure 2 we show the proportion of countries reporting government as the primary funder of different elements of IMCI in first-level health facilities by under-five mortality level.

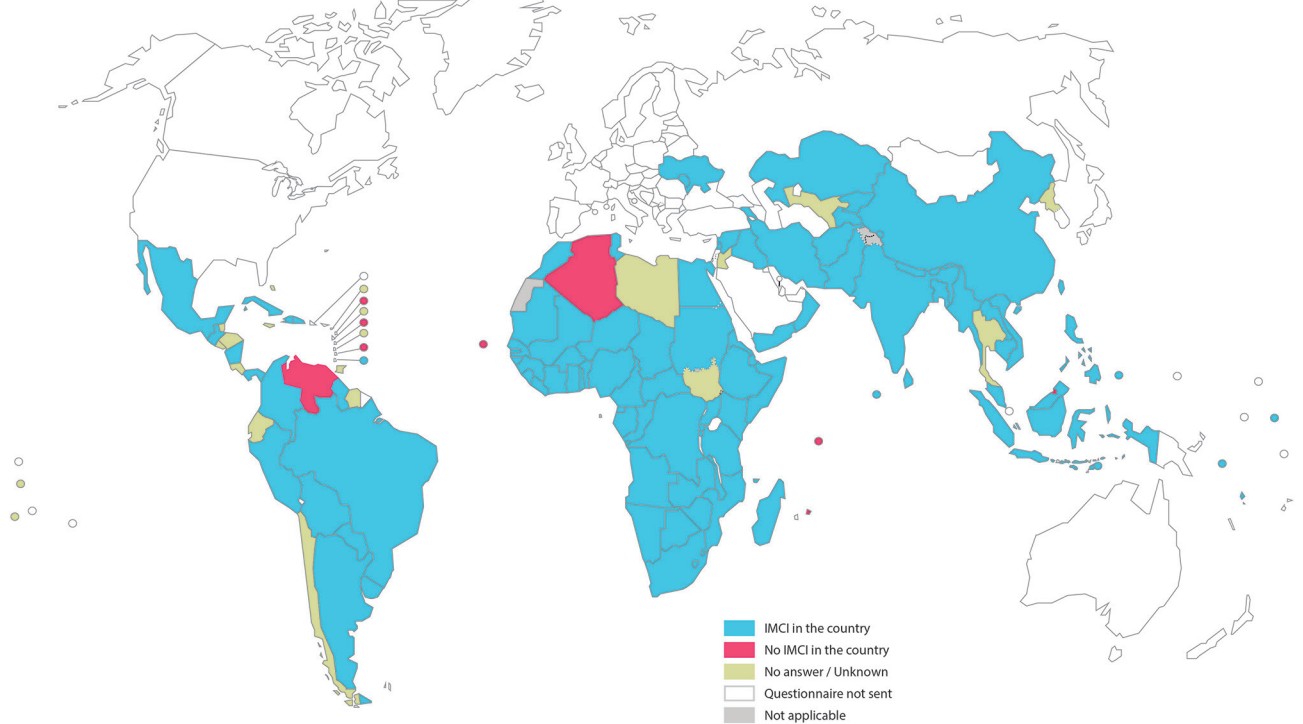

**Figure 1** Survey summary map. IMCI, Integrated Management of Childhood Illness.

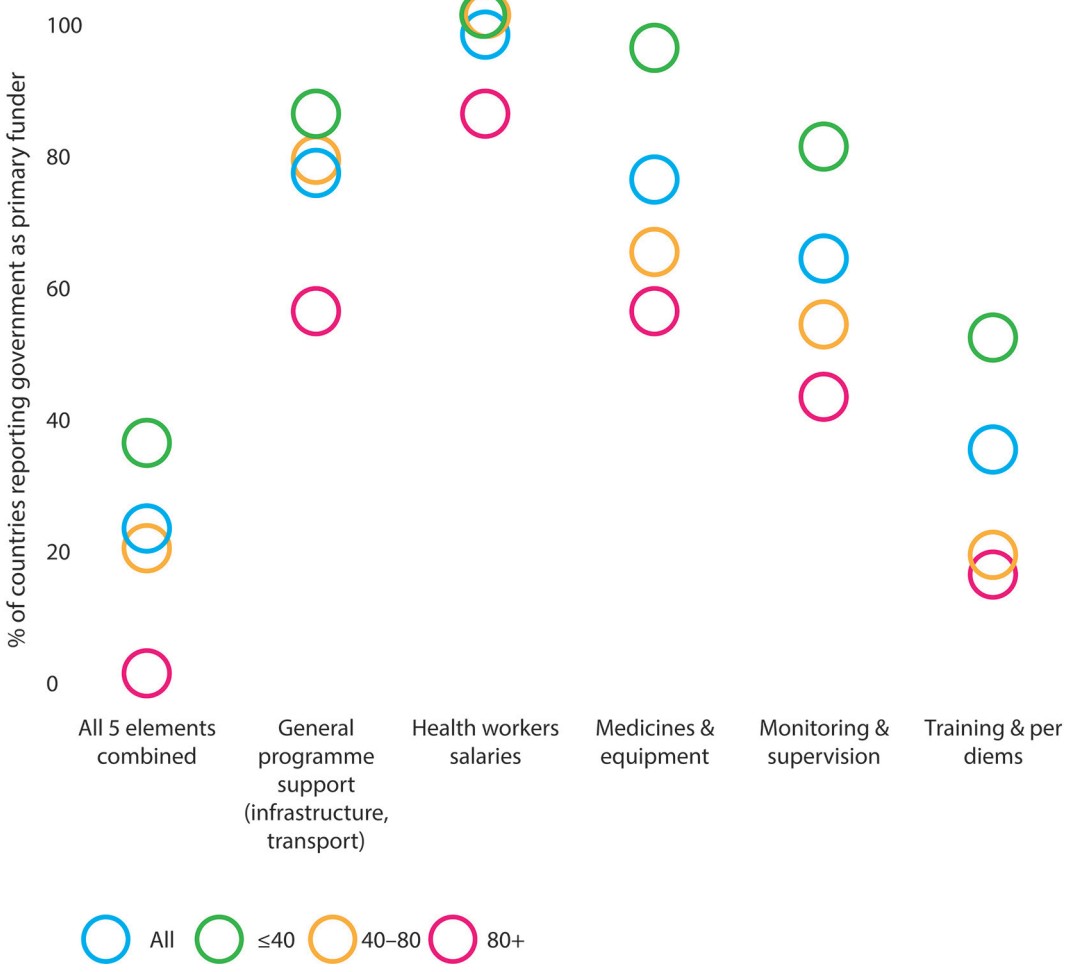

**Figure 2** Proportion of countries reporting government as primary funder of various elements of Integrated Management of Childhood Illness in first-level health facility by under-five mortality level.

## Main results

For the purpose of the survey, IMCI is said to be implemented if it is included in national and district work plans and budgets, if district health facilities use the strategy to provide care for the sick child (with or without community-based IMCI), and if providers are trained in IMCI at national and district levels. Of the 95 countries included in the full analysis, 81 responded to the question on the proportion of districts implementing IMCI. Of these 81 responding countries, 51 (63%) reported implementing IMCI in all districts.

Figure 3 shows the proportion of countries reporting the components of IMCI they have implemented. While 92 (98%) of the 94 responding countries reported having implemented the first component of IMCI (improving health worker skills), 89 (95%) reported implementing the second (strengthening health systems) and 78 (83%) responded having implemented the third (improving family and community practices). Overall, 76 (81%) of the 94 countries reported implementing all three components.

Forty-four countries reported implementing IMCI in more than 90% of districts and having all three IMCI components in place; these are considered full-implementer countries. These 44 full-implementer countries were 3.6 (95% CI 1.5 to 8.9) times more likely to achieve MDG-4 than other (not full implementer) countries, and are home to 160 million (24%) of the world's under-five children.

### First component: improving health worker skills

The original training course for health workers lasted 11 days, with 30% of time spent in clinical sessions.[12] Most responding countries (42%; 39/92) used shortened or abridged versions (ranging from 6 to 8 days) of the original 11-day course. Clinical practice was at least 25% of the training time in 79% of the countries (72/91). Nearly three-quarters of the countries (72%; 67/93) reported implementing some sort of IMCI preservice training.

Generic case management guidelines and related training materials were developed with the aim of being appropriate in LMICs where U5MR is higher than 40 per 1000 live births, and where there is transmission of *Plasmodium falciparum* malaria.[13] Over the years, IMCI chart booklets have been continuously adapted to the context and changing scenarios of each country. Since the last global update in 2014, 81% (69/85) of countries reported having updated their national chart booklets. Recognising

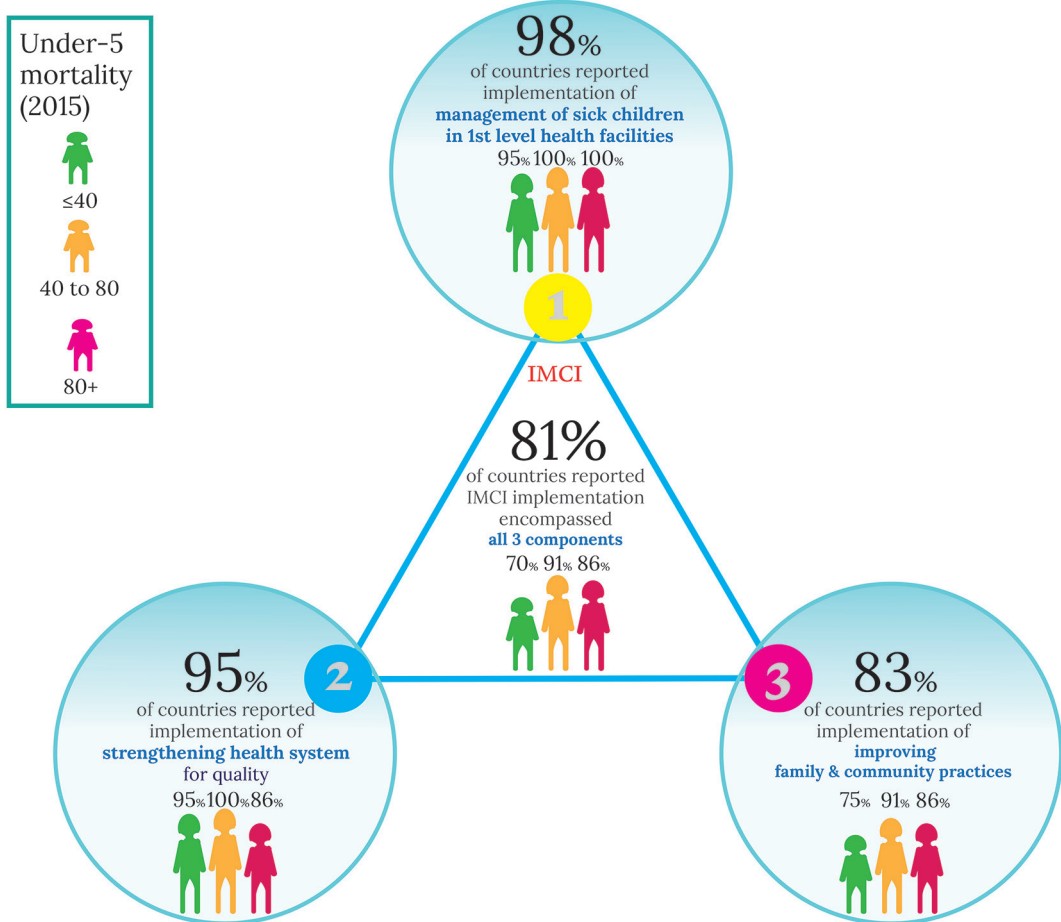

**Figure 3** Implementation of the three IMCI components. IMCI, Integrated Management of Childhood Illness.

the increased proportional burden of newborn mortality, nearly all countries (95%; 89/94) adapted IMCI guidelines to include care of the sick newborn in the first week of life. Development or revision of national treatment guidelines based on the 2013 revision of the WHO Pocket Book of Hospital Care for Children and its adaptation in referral facilities were reported by 55% (48/87) and 44% (39/88) of responding countries, respectively. Considering Emergency Triage Assessment and Treatment, 23% of 65 responding countries reported introducing these guidelines in at least 75% of hospitals. This proportion is lower (17%) in the 41 middle-mortality and high-mortality countries than in the 24 low-mortality countries (38%).

The most common conditions added to the guidelines in national adaptations were jaundice in newborn babies and young infants (77%; 72/93), HIV (63%; 59/93), sore throat (59%; 55/93) and skin conditions (44%; 41/93). Other illnesses frequently included are tuberculosis and dengue haemorrhagic fever. Adaptations varied according to regional or national epidemiological profile.

### Second component: strengthening health systems
Improving and sustaining the quality of care (QoC) is an integral part of strengthening health systems for paediatric patients. An assessment of quality of

paediatric hospital care was conducted in 42% (39/92) of responding countries. Fifty-two per cent (34/65) of middle-implementer and high-implementer countries reported having such assessment compared with only 8% (1/13) of low-implementer countries. Fifty-eight per cent of the 92 responding countries reported having a paediatric QoC improvement programme for health facilities in their MoH. Countries that report improvement programmes together account for nearly 3 million (50%) of all 5.9 million under-five deaths occurring worldwide in 2015.

Supervision and monitoring are key to sustainable QoC and to strengthening the overall health system. Few countries (15%; 10/66) reported that more than 75% of first-level facilities received at least one supervisory visit in the last 6 months; half (33/66) had less than one-quarter of these facilities receiving visits. In general, monitoring of IMCI implementation has been carried out infrequently. Only 33% of countries (30/91) reported having a comprehensive IMCI monitoring and evaluation plan. Fifteen per cent (3/20) of high-mortality countries reported having such a plan, while this proportion was 38% (27/71) in middle-mortality and low-mortality countries. Still, more than two-thirds of countries (70%; 66/94) reported that their health management information system (HMIS)

includes monitoring indicators for IMCI. Most of these indicators are collected at the primary healthcare level (83%; 54/65); only four countries reported having monitoring indicators for IMCI collected at community level.

In 2009, WHO launched the 'Managing Programmes to Improve Child Health' course which has been introduced in almost half (48%; 42/88) of the responding countries. The number of managers trained in this course per under-five population varied widely, ranging from 1 manager per 840 000 under-five children in Lao PDR to 1 manager for 1500 children in Kiribati—a >500-fold difference. Tools for bottleneck analysis and strategic planning have been introduced in 33 of the 79 responding countries (42%).

The main innovations reported by countries are related to information communication technologies which include eHealth, mHealth and Rapid SMS. Use of such innovations in the implementation of IMCI was reported over six categories: data collection and reporting (70%; 60/86), supply chain management (58%; 51/88), behaviour change communication (54%; 48/89), health provider training and education (54%; 46/85), electronic decision support (49%; 42/85) and financial transactions and incentives (41%; 35/86).

### Third component: improving family and community practices

Improving family and community practices was included as part of IMCI implementation by 83% (78/94) of responding countries (figure 3). Eighty-nine per cent (48/54) of medium-mortality and high-mortality countries report implementation of this component, while 75% (30/40) of low-mortality countries reported implementing it. In general, activities to *promote key family practices for child health* have been implemented through home visits, social mobilisation or community groups. The main delivery mechanisms used are illustrated in figure 4. The majority of responding countries (71%; 45/63) reported that one CHW covered up to 1000 under-five population. Supervision for CHWs was mainly provided by IMCI-trained supervisor (63%; 58/92) and through mentorship in health facilities (45%; 41/92).

*Community care for children under 5 years of age* was reported to be provided by more than three-quarters (78%, 72/92) of responding countries. The proportion of countries with CHWs providing treatment for children, in general, was higher in high-mortality (85%; 17/20) and medium-mortality (94%, 31/33) countries compared with low-mortality countries (62%, 24/39). These variations are similar to those observed for the presence of community case management (CCM) policy in countries. Among the 72 countries that reported that CHWs provide care for children, 93% (67 countries) reported having CCM for diarrhoea. Of these, 80% (53/66) reported having the relevant written policy in place (one country did not respond). Conversely, six countries reported having the policy in place but not implementing it. Among the 72 countries where CHWs provided treatment for children, only 74% (53/72) reported that CHWs provide CCM for

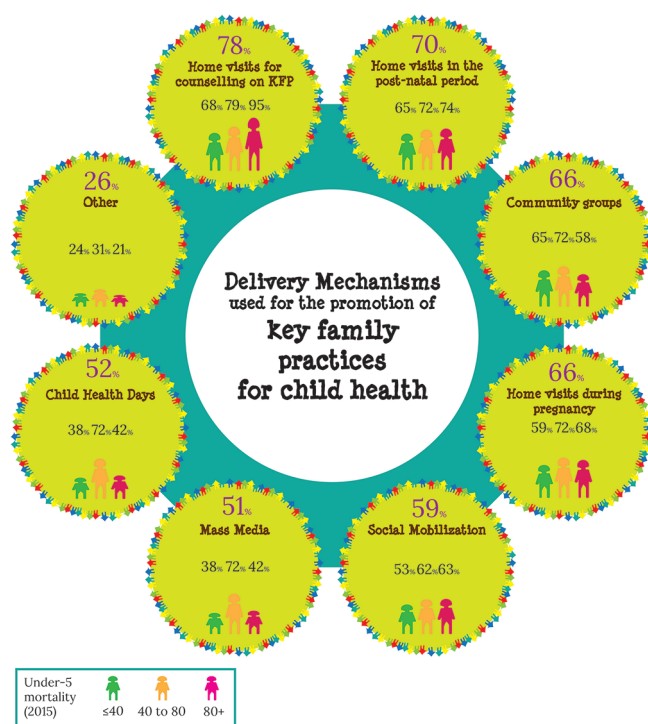

**Figure 4** Main delivery mechanisms used for the promotion of key family practices related to child health.

pneumonia. From these 53 countries, 43 (81%) reported having the relevant written policy in place. In 2012, the key elements of CCM were brought together in a package—iCCM,[14] typically delivered by CHWs at the community level. It encompasses treatment for (1) childhood pneumonia with antibiotics, (2) diarrhoea with zinc and oral rehydration salts and (3) malaria with artemisinin combination therapy. Provision of iCCM was reported by 72% (52/72) of these responding countries.

Other aspects of community care, also supported by WHO and Unicef, include the management of severe acute malnutrition (SAM)[15] and home visits.[16] CCM for SAM was mentioned by 85% (61/72) of countries. Of the 95 countries participating in the survey, 65 reported that CHWs provide care for newborn babies. Of these countries, 91% (59/65) reported providing home visits to newborns. In comparison, 76% (55/72) provided home visits for children beyond the newborn period.

### Strengths, barriers and ways forward

The major strengths of IMCI most frequently identified by 93 responding countries were the holistic approach to the child (90%; 84 countries), the rational use of medicine (89%; 83 countries), the quality of health services (87%; 81 countries) and the efficiency of service provision (80%; 74 countries).

At national level, the reported major barriers to implementing IMCI were budget for training (85%; 78/92), mentorship and supervision (74%; 68/92), cost or sustainability of activities (63%; 58/92) and availability of a dedicated budget line (60%; 55/92). Major barriers reported at regional/district level were staff turnover (84%;

76/91), budget for training (82%; 75/91) and mentorship and supervision (74%; 67/91). Barriers reported at facility level were staff retention (80%; 73/91), mentorship and supervision (79%; 72/91), and staff motivation (74%; 67/91). High-mortality and/or low-income countries were more likely to report barriers related to budget for training, and medicine procurement and supply chain at national level. High-implementer countries were more likely to report issues around mentorship and supervision, and less likely to report barriers on programme management and on political support and ownership.

When asked to suggest ways to strengthen child health services and make IMCI more relevant to today's needs, countries commonly advocated for better integration of current strategies into general health services, with a costed work plan and committed budget, and improved training at all levels, including preservice training, and supportive supervision (helping to make things work, rather than checking to see if something went wrong) and mentoring. Health systems issues were said to limit implementation, with a strong need for a functioning supply chain, HMIS and monitoring and evaluation system. Better use of communications technologies was also suggested.

At the policy level, strategic planning, improved internal coordination (via a dedicated focal person for child health, regular review meetings, national steering committee) and effective coordination with partners to overcome fragmentation would be essential. Respondents said compliance with national guidelines and good implementation practices should also be recognised and rewarded. Community-level activities including outreach and home visits need more emphasis, as do the management of sick newborns at all levels. To achieve these goals, high-level advocacy and political support is said to be required, ensuring government leadership and ownership.

## DISCUSSION

This study has some limitations, most of which are inherent to survey design. Respondents may not feel comfortable providing answers that present the country in an unfavourable light and therefore may not provide accurate answers. Also, respondents may not always be fully aware of the situation in the country. To minimise this limitation, we provided clear instructions for questionnaires to be filled out by a three-person in-country team, but only five countries followed this instruction. Another 16 countries had questions answered by a two-respondent team. Bias due to non-response may also have occurred as respondents who chose to answer certain questions may differ from those who chose not to respond. Finally, the WHO African region is likely to be over-represented in the survey, representing 43% of the responding countries, while the European region represents 6% of them. Disaggregation by mortality level and income groups reduces the effect of this over-representation on the interpretation of results. Despite these limitations, this survey provides a unique opportunity to better understand how and in which direction the implementation of IMCI has evolved over the 20 years since its inception. It is also a unique opportunity to help steer the direction of future child health strategies.

This article summarises the first survey assessment of IMCI implementation status in 95 responding countries worldwide, 20 years after its launch in the mid-1990s.[11] In 1998, 12 countries had already moved into the expansion phase of IMCI's first component and had begun introducing the other two components.[17] Twenty years later, coverage of IMCI is reported to be comprehensive in many target countries; however, few countries have achieved full scale-up and, in many aspects, implementation remains incomplete. Moreover, regardless of high reported implementation, the strategy is still not reaching the children who need it most, since coverage is lowest in high-mortality countries. Nevertheless, it is worth noting that high-implementer countries (≥90% of districts) with all three IMCI components in place are home to 160 million (24%) of the global under-five children. These full-implementer countries were 3.6 (95% CI 1.5 to 8.9) times more likely to achieve MDG-4 than other (not full implementer) countries. Also of note is the fact that among the 24 low-income and lower-middle income countries that achieved the MDG-4 target, 18 (75%) are full-implementer countries. Our results reinforce the original concept that full implementation of IMCI can lead to a substantial impact on child health and survival.

The training course for health workers developed by WHO and Unicef is one of the key elements of IMCI strategy. Some limited evidence suggests the original 11-day training course to be more effective than shortened training[18]; however, concerns have been raised regarding the costs and the absence of health workers from their duties.[19] Most countries reported using shortened or abridged versions of the original course. Flexibility to add conditions according to regional and national epidemiological profile has facilitated adaptation of IMCI guidelines to a changing epidemiological profile. Supervision is the weakest area under the second IMCI component, with only 15% of countries reporting more than three-quarters of first-level facilities receiving at least one supervisory visit in the 6 months before the survey. Similar results have recently been shown in a study carried out in Afghanistan, where only 11% of health workers received supervision during the past 6 months.[20] Lack of supervision was also reported as a barrier in an overview of IMCI implementation in West Java province, Indonesia.[21] The most commonly mentioned barriers must be foremost in the minds of our thinking on and redesigning the future of child health and development.

There is widespread recognition that IMCI will only result in improvements in child health and survival if training activities are accompanied by effective efforts to strengthen health systems and reach children and mothers in the community. It has also been recognised that 'strengthening health systems is the best way to safeguard against health crises'.[22] Sadly, results highlight the weakness of the

implementation of IMCI second component. Supervision and mentorship were among the areas with the least impressive results. Special attention should be directed to the second component of IMCI. The sustainability of IMCI, and of the gains in child health this strategy can help achieve, will depend on countries' efforts to make real improvements in their health systems—beginning with providing adequate funding for health worker salaries, training and supplies, among other inputs.

## CONCLUSION

Despite the multiple and extensive constraints in IMCI implementation reported here, the findings of this survey also provide an indication of the value and esteem that country-level stakeholders assign to IMCI. Results point to a unique opportunity to help steer future policies, programmes and strategies to promote child health. Given the many competing priorities of survey respondents, the 80% response rate obtained reveals the interest IMCI still elicits, especially in LMICs, and suggests that a strengthened IMCI has a role in attending the call for 'Survive, Thrive, Transform' from the Global Strategy for Women's, Children's and Adolescents' Health 2016–2030.[23] IMCI has generated a paradigm shift towards integrated programming for child health globally and in countries which should be built on in the current era of the Sustainable Development Goals (SDGs), under which global leaders have committed to end preventable newborn and child mortality. Full implementation of IMCI in health facilities and communities with a critical focus on health systems strengthening will be decisive for countries to secure universal healthcare and help achieve the health-related SDGs.

**Author affiliations**
[1]Department of Maternal Newborn Child and Adolescent Heath, World Health Organization, Geneva, Switzerland
[2]Instituto de Saúde Coletiva, Universidade Federal Fluminense, Niteroi, Brazil
[3]The Global Fund to Fight AIDS, Geneva, Switzerland
[4]UNICEF, New York, USA
[5]World Health Organization, Regional Office for Africa, Ouagadougou, Burkina Faso
[6]World Health Organization, Regional Office for East and Southern Africa, Harare, Zimbabwe
[7]World Health Organization, Regional Office for Africa, Brazzaville, Congo
[8]World Health Organization, Regional Office for the Americas, Washington, USA
[9]World Health Organization, Regional Office for Eastern Mediterranean, Cairo, Egypt
[10]World Health Organization, Regional Office for Europe, Copenhagen, Denmark
[11]World Health Organization, Regional Office for South-East Asia, New Delhi, India

**Acknowledgements** The authors thank their colleagues from WHO and Unicef Country Offices and those from the Ministries of Health for their contribution on responding to the survey questionnaires and on providing follow-up of the survey.

**Contributors** The authors bring a combination of expertise in epidemiology, demography, statistics, child health interventions and policies, public health and advocacy. CB-P conceptualised and wrote the first drafts of the paper. GL, TRD and NO performed the analysis. GL conceptualised and performed the illustrations and revised the first draft. SLD analysed the qualitative information and wrote the last section of the paper (Strengths, barriers and ways forward). BD and SA developed the questionnaire. SA revised several versions of the paper, providing advice on IMCI-specific issues, helped coordinate the distribution and application of questionnaires and followed up with WHO Regional Offices. OAA-P, TD, PH, BB-R,

JA-R, KS, AK, MW, RM and NR organised, tested, followed up and reviewed the questionnaires sent to countries. All authors contributed intellectual content and critical revisions to the manuscript. All authors approved the final manuscript. CB-P is the guarantor.

**Funding** The Strategic Review of IMCI and iCCM was funded by the Bill and Melinda Gates Foundation; the authors donated their time to writing the article; the publication fees for the supplement were funded by the Health Systems Research Unit, South African Medical Research Council.

**Disclaimer** The authors are staff members of WHO. The authors alone are responsible for the views expressed in this publication and they do not necessarily represent the decisions or policies of WHO.

**Competing interests** None declared.

**Patient consent** Not required.

**Provenance and peer review** Not commissioned; externally peer reviewed.

**Data sharing statement** All data will be available in the public domain (IMCI Global Implementation Survey Report) at: http://who.int/maternal_child_adolescent/documents/imci-global-survey-report/

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
