## [Reviewer comments · BMJ Open]

ARTICLE DETAILS

TITLE (PROVISIONAL)	Global implementation survey of the Integrated Management of Childhood Illness (IMCI) – twenty years on
AUTHORS	Boschi-Pinto, Cynthia; Labadie, Guilhem; Dilip, Thandassery; Oliphant, Nicholas; Dalglish, Sarah; Aboubaker, Samira; AGBODJAN-PRINCE, Olga; Desta, Teshome; Habimana, Phaniel; Butrón-Riveros, Betzabé; Al-Raiiby, Jamela; Siddeeg, Khalid; Kuttumuratova, Aigul; Weber, Martin; Mehta, Rajesh; Neena, Raina; DAELMANS, Bernadette; Diaz, Theresa

VERSION 1 – REVIEW

REVIEWER	Chandrashekhar T Sreeramareddy International Medical University Kuala Lumpur Malaysia
REVIEW RETURNED	27-Aug-2017

GENERAL COMMENTS	This article addresses an important issue of child in the view of post-MDG, SDG era child health targets. The survey done by WHO is an important one for further directions towards this program, but the article lack important short comings in background and methods, for this to be a published as a scientific article. My comments appended a annotation in the attached PDF. Results are too boring to read through percentages and fractions, would the authors list some countries which are worst or best performers, and these also be discussed in the last section. Could the authors do some analyses for SSA countries and other LMICs, or by World bank taxanomy, etc. Were the representatives of health ministry or WHO or UNICEF anonymous, was their position, qualification, level of involvement in IMCI was available, then be presented here for assessing credibility of information given. Can some comparison be made with existing document about IMCI in these countries. There is not sufficient literature cited in discussion and lack of any relevant citation in background, i have not commented on discussion section Lastly, the discussion ends very abruptly, after limitations, without any summary or conclusion/s. - The reviewer also provided a marked copy with additional comments. Please contact the publisher for full details.
---

REVIEWER	Ghulam Farooq Mansoor FHI360 Afghanistan
REVIEW RETURNED	05-Sep-2017

GENERAL COMMENTS	This is an interesting study and provide very useful information on how the IMCI has evolved since its inception in the 90s. some minor comments have been embedded in the document for the authors to address - The reviewer also provided a marked copy with additional comments. Please contact the publisher for full details.c
---

REVIEWER	Tanya Doherty South African Medical Research Council, South Africa
REVIEW RETURNED	05-Sep-2017

GENERAL COMMENTS	Thank you for the opportunity to review this paper reporting on findings from a global implementation survey of IMCI. The article presents important findings on the current state of implementation of IMCI across the globe. My main comments on the article relate to the methods section which is rather thin on details. My detailed comments are outline below:  • In the abstract the authors could list outcome measures such as 'prevalence of IMCI implementation stratified according to the three components' • The introduction is very short (one paragraph) and should be extended to provide a more thorough background to the survey, citing relevant literature on IMCI implementation and providing the justification for the survey. • The methods section need to be more detailed and would benefit from sub-headings e.g. study design, sampling, data collection, data analysis. • Please specify how the sampling frame of countries was developed, what kind of questionnaire was developed (online survey etc), how were questionnaires distributed and returned? • What were the procedures for data cleaning? Were countries contacted for missing information? • Under data analysis please specify how this was performed. Simple frequencies and means? How was open-ended data handled? • Figure 3 is missing from the manuscript file, only the figure title is included. • When referring to figure 4 on page 5 please give proportions as well since that is what is shown in the figure. • Page 5, line 22- use LMIC not 'developing countries'. • Page 6, line 57, please add a reference for the statement "these variations are similar to..." • When reporting findings from open-ended questions do not use the term 'respondents said' since this data is not from interviews. You could rather use 'in-country teams reported' • Page 8, line 25, there is a repetition of 'the strategy is not..' • Please add to the limitations that these were self-administered questionnaires and the researchers did not verify the information reported by countries. Since the survey was developed by WHO and UNICEF, the agencies who developed the IMCI strategy, there is a risk that countries may report data that reflects a better picture of IMCI implementation than the reality.
--

VERSION 1 – AUTHOR RESPONSE

Responses to Reviewers

Page references are related to the unmarked document

Reviewer(s)' Comments to Author and Responses:

Editorial Requests

- Please revise your title so that it includes your study design. This is the preferred format for the journal.

Title revised to: “Global implementation survey of the Integrated Management of Childhood Illness – twenty years on”

Title; Page 1

- The introduction section is currently far too short. Please expand this section to summarize the background literature on this topic and the rationale for carrying out the study.

The main reason why we had a deliberately short introduction was because this paper is part of a series in which the first paper deals with the background of IMCI. The second reason was the full length of the paper. We have now expanded it, according to the suggestions.

Introduction; Page 3

- The methods section is also not reported in enough detail. For example, how were teams contacted/ recruited? What was the sampling strategy? Were there any inclusion criteria? How did you analyse the data? The methods should be reported in enough detail so that other researchers can reproduce your work. Readers should not have to refer to reference 6. A copy of the questionnaire should be included as a supplementary file.

Again, the main reason for a short Methods section was the full length of the paper. The section has been significantly expanded and more details on sampling strategy/inclusion criteria, data analysis are now provided according to STROBE and reviewers' and editors' suggestions.

Methods; Pages 3 and 4

An additional file is now included with the English version of the questionnaire.

Reviewer: 1

Reviewer Name: Chandrashekhar T Sreeramareddy

Institution and Country: International Medical University, Kuala Lumpur, Malaysia

Abstract

1.Objectives

Please write the survey objectives here.

The survey objectives are now described.

Abstract; Page 1

2.Method

Specify that this was a cross-sectional self-administered, questionnaire survey.

This is now specified.

Abstract; Page 1

3.Participants

This should be representatives of the implementing organisation in the countries.

This has been modified to describe respondents
Abstract; Page 1

4.Intervention

Could delete the section sub heading.
This has been deleted

5.Discussion

Change to Conclusion.
This has been done
Abstract; Page 2

Summary box

First sentence irrelevant remove
First sentence has been removed
Page 2

Introduction

A brief introduction about IMCI and its evolution, implementation, coverage, effectiveness, impact etc. may be briefed here.
We have added a couple of paragraphs to the introduction
Page 3

Cite them here, briefly review them, what they report, what they don't; how this would fill that gap.
We have added some further citation to the introduction
Page 3

Methods

Please structure this section as per the STROBE check list heading under methods, design, participants, settings are sufficiently described.
We have structured the session according to STROBE check list, although not always in the exactly same order
Pages 3 and 4

Patient involvement- Delete that.
This is a requirement of the Journal

My comments appended a annotation in the attached PDF. Results are too boring to read through percentages and fractions, would the authors list some countries which are worst or best performers, and these also be discussed in the last section. Could the authors do
This paper is supposed to present the general findings of the IMCI global implementation survey. We have provided some results according to mortality rates and income levels as well as some brief discussion on the full implementer countries. However, due to the already long length of the paper we have not been able to present more disaggregated information. These are presented in the full report (http://www.who.int/maternal_child_adolescent/documents/imci-global-survey-report/en/)

We agree the paper is heavy on results. We have tried to suppress some information and slightly shorten it where possible

Were the representatives of health ministry or WHO or UNICEF anonymous, was their position, qualification, level of involvement in IMCI was available, then be presented here for assessing credibility of information given. Can some comparison be made with existing document about IMCI in these countries?

There were about 200 respondents with diverse background. We have included a general statement on the majority of respondents:

“Respondents comprised in-country teams consisting of a combination of representatives of the ministry of health (MoH) and country offices of WHO and UNICEF or in some circumstance just representatives of the MoH. Respondents were IMCI experts with country experience”.

Methods; page 4

There is not sufficient literature cited in discussion and lack of any relevant citation in background, i have not commented on discussion section. Lastly, the discussion ends very abruptly, after limitations, without any summary or conclusion/s.

We have included some more literature.

The discussion did not end after limitations. The following paragraph closed the section: “Despite these limitations, this survey is a unique opportunity to better understand how and in which direction the implementation of IMCI has evolved over 20 years since its inception. It is also a unique opportunity to help steer future child health directions. The 80% response rate obtained reveals the interest IMCI still elicits, especially in LMICs, and is an indicator that it should be strengthened and properly re-designed to attend the call for “Survive, Thrive, Transform” from the Global Strategy for Women’s Children’s and Adolescents’ Health 2016-2030 (16). The 95 countries that participated in this Global Survey are home to 82% of the global under-five population and account for 95% of the 5.9 million deaths that occurred among children less than five years of age in 2015. These results are therefore extremely informative and should be used to assist in formulating strategies, policies and activities to support improvements in health and survival of children and to help achieving the UN health-related, post-2015 Sustainable Development Goals (SDGs).”

However, we have slightly modified it to make it more focused on implementation aspects. The last paragraph now reads:

“Despite the multiple and extensive constraints in IMCI implementation reported here, the findings of this survey provide an indication of the value and esteem that country-level stakeholders assign to IMCI. Results point to a unique opportunity to help steer future policies, programmes and strategies. Given the many competing priorities of survey respondents, the 80% response rate obtained reveals the interest IMCI still elicits, especially in low and middle income countries, and suggests a strengthened IMCI has a role to attend the call for “Survive, Thrive, Transform” from the Global Strategy for Women’s Children’s and Adolescents’ Health 2016-2030 (21). Truly, IMCI has generated a paradigm shift towards integrated programming for child health globally and in countries. This must

be built upon in the current era of the sustainable development goals (SDGs), when global leaders have committed to reach a grand convergence in child health and end preventable newborn and child mortality. Full implementation of IMCI in health facilities and communities with a critical focus on health system strengthening and on emergency crises will therefore be decisive for countries to secure universal health care and to help achieve the UN health-related, post-2015 SDGs.”

Reviewer: 2

Reviewer Name: Ghulam Farooq Mansoor

Institution and Country: FHI360, Afghanistan

Methods

Is this Integrated Community Case Management? Please spell out with first use.

This is spelled out the first time in the introduction, perhaps this was missed by the reviewer.

The provision of timely and effective integrated treatment of diarrhoea, pneumonia and malaria – integrated community case management (iCCM) (11) – by CHW to populations with limited access to facility-based health care providers was reported by 64% (46/72) of responding countries. Among the 66 malaria-endemic countries that responded to this question, the same proportion (64%; 42/66) reported existence of iCCM. Policies were present in 34 of these 42 countries (81%). Please note that we have slightly modified this paragraph, but iCCM is still spelled out.

Page 7

Not sure if district is a right comparable unit. Districts sizes may vary from one country to another. Need some further elaboration on this.

Although population size may vary, district is the most common health management unit in many of the countries surveys and we feel

it is the best representation of the extent of scale up of IMCI.

Results

Figure 3 is not visible

Figure 3 was provided and we do not know the reason why it is not visible, but will upload it again

Suggest adding explanation/definition for the supportive supervision in the IMCI context. Specifically did the supportive supervision include.

Supportive supervision is generally done using a check list the supervisor observes the care provided, praises the provider for things

that are done well and discusses weaknesses followed by demonstrating how to correctly administer care. It also entails that the

supervisor ensures the provider has all necessary equipment and tools to provide patient care.

Although respondents suggested

improving supportive supervision none of them described in detail what this supervision included, assuming that this was well known.

So we feel we cannot detail specifically what was done. Instead we included the following in brackets (“helping to make things work,

rather than checking to see if something went wrong”).

Results; Page 8

“Of the 95 countries included in the full analysis, 81 responded to the question on the proportion of districts implementing IMCI: 51 (63%) countries reported implementing IMCI in all districts; 54 (67%) in 90% or more districts and 62 (77%) countries reported implementing it in at least three quarters of their districts”...

This is confusing; how is this overlap in responses possible? 63% countries reported implementing IMCI in all districts then why 67% say in 90% or more districts and 77% in at least 3 quarters of districts. please elaborate.

These are not mutually exclusive categories. However, we have changed and shortened the sentence to read as follows: “Of the 95

countries included in the full analysis, 81 responded to the question on the proportion of districts implementing IMC of which 51 (63%)

countries reported implementing IMCI in all districts.”

Main Results; Page 5

Suggest adding a sentence on the number of countries implementing all three components.

We have added the following: “Overall 76 of the 94 countries (81%) reported implementing all three components.”

Main Results; Page 5

“Most responding countries (42%; 39/92) used shortened or abridged versions of the original course.” Spell out the length of the abridged version.

We have added that abridged training ranges from 6 to 8 days: “Most responding countries (42%; 39/92) used shortened or abridged

versions (ranging from 6 to 8 days) of the original 11-day course.”

First component; Page 6

“From these 53 countries, 43 reported having the relevant written policy in place. Of the 66 malaria-endemic countries that responded to the question, 89% (59/66) reported CCM for malaria by CHWs; all 27 low income countries implemented it.” Please add percentages.

This has been done

Third component; Page 7

Discussion

“Moreover, despite the high reported implementation rates, the strategy is still not reaching the strategy is not yet reaching the children who need it most - coverage of IMCI is lowest in high mortality countries.” Redundancy please check.

We thank the reviewer for noting this mistake. This has been fixed

“Our results show a higher proportion of countries with an IMCI focal point at national level than at the sub-national (regional or district) level. How is the working group/task force?

We did not ask about working groups or task forces and are therefore not able to include this

Reviewer: 3

Reviewer Name: Tanya Doherty

Institution and Country: South African Medical Research Council, South Africa

In the abstract the authors could list outcome measures such as ‘prevalence of IMCI implementation stratified according to the three components’

Reviewer 1 has asked for deletion of this item. Because this study is not a trial, we agreed to delete it

The introduction is very short (one paragraph) and should be extended to provide a more thorough background to the survey, citing relevant literature on IMCI implementation and providing the justification for the survey.

As mentioned before, the main reason why we had a deliberately short introduction was because this paper is part of a series in which the first paper deals with the background of IMCI. The second reason was the full length of the paper. We have now expanded it, according to the various suggestions.

Introduction; Page 3

The methods section need to be more detailed and would benefit from sub-headings e.g. study design, sampling, data collection, data analysis.

Please specify how the sampling frame of countries was developed, what kind of questionnaire was developed (online survey etc), how were questionnaires distributed and returned?

What were the procedures for data cleaning? Were countries contacted for missing information?

Under data analysis please specify how this was performed. Simple frequencies and means? How was open-ended data handled?

This has now been done according to the STROBE checklist as also suggested by Reviewer 1. We have added the details requested

and will also add the questionnaire as a supplementary file

Methods; Pages 3 and 4

Figure 3 is missing from the manuscript file, only the figure title is included.

Figure 3 was provided and we do not know the reason why it is not visible, but will upload it again

When referring to figure 4 on page 5 please give proportions as well since that is what is shown in the figure

This has been done

Results; Page 5

Page 5, line 22- use LMIC not 'developing countries'.

This has been done

Page 6, line 57, please add a reference for the statement "these variations are similar to..."

This sentence refers to the results of the survey; variations shown for the implementation of CCM by mortality levels were similar to

those observed on CCM policy. It is not referring to other studies from the literature. We have rephrased it to make it more clear.

When reporting findings from open-ended questions do not use the term 'respondents said' since this data is not from interviews. You could rather use 'in-country teams reported'

This has been changed. The paragraphs have been slightly changed.

Pages 7 and 8

Page 8, line 25, there is a repetition of 'the strategy is not..'

We thank the reviewer for noting this mistake. It has been corrected

Please add to the limitations that these were self-administered questionnaires and the researchers did not verify the information reported by countries. Since the survey was developed by WHO and UNICEF, the agencies who developed the IMCI strategy, there is a risk that countries may report data that reflects a better picture of IMCI implementation than the reality.

When researchers suspected that the information was not correct, they went back to the countries to double checking. When

needed, responses were corrected. please note we have the following:

“Responses were tracked by WHO regional offices to maximize return and clarify inconsistencies or omissions.”

“Clarification was sought from countries with missing data, of which some responded”.

Methods; Page 4

We have mentioned in the paragraph on limitations (Conclusions; page 9) that: “... respondents may not feel comfortable providing

answers that present the country in an unfavourable manner and therefore may not provide accurate answers. Also, respondents

may not always be fully aware of the situation in the country.”

VERSION 2 – REVIEW

REVIEWER	Tanya Doherty South African Medical Research Council, South Africa
REVIEW RETURNED	25-Oct-2017

GENERAL COMMENTS	Thank you for the opportunity to review a revision of this manuscript. The authors have addressed the reviewer comments well. There are just a few areas where I feel some minor changes are necessary:  • I do still believe that the methods section would benefit from some sub headings. Currently it is quite dense with a lot of information and long paragraphs. It would help readers to split this information into sub headings such as: study design, sampling, data collection, data analysis. Etc. • The statement at the end of the methods section “as is common with self-administered...” belongs in the limitations section of the discussion. • The limitations section should come before the conclusion so that the conclusion provides the main messages and concluding thoughts only. • The authors present a new finding in the abstract and discussion: “These full implementer countries were 3.6 [95% CI 1.5 8.9] times more likely to achieve MDG4 than other (not full implementer) countries.” This result does not appear anywhere in the results section, only in the abstract and conclusion. Also the analysis methods used to generate this estimate are not included in the data analysis section. Either this result needs to be included in the results section and the analysis methods e.g. logistic regression or whatever method was used needs to be described in the data analysis section or it should be removed. • The issue of IMCI in emergency settings is raised in the conclusion but emergency settings are not mentioned in the results. Could the authors bring our results relevant to emergency settings in the results section so that readers can link the concluding recommendations with the study results. • It is surprising that sustainability is not mentioned in the conclusions given the strong finding in figure two that high under-5 mortality countries had the least government contribution to the five elements of IMCI. • The statement in the conclusion: “Our findings show that low
---

	income countries particularly hailed IMCI efficiency in programming and in service provision, as well as equity in access and coverage of intervention” is unclear. Please can this be re phrased. • There are some typos and grammatical errors in the revised text which need a careful edit.
--	--

REVIEWER	Chandrashekhar T Sreeramareddy International Medical University, Malaysia
REVIEW RETURNED	02-Nov-2017

GENERAL COMMENTS	The revised MS adheres to the PRISMA and technical language confirms with the research paper. A few minor issues Methods: questionnaire was pretested, how was the questionnaire in six UN languages validated. Need not use regional office (RO), may confuse with responsible officers. "As is common with self-administered questionnaires respondents may have inaccurately reported the level and extent of IMCI implementation, most likely reporting a more “positive” scenario than the reality." should be written under discussion, as limitations. RESULTS: Eighty-nine per cent (82/94) when the number analysed was just 94, please have an alternative methods for presenting such number less than 100 where % do not make much sense. conclusion paragraph is way too long, most of which are implications or ways forward from this study findings. there needs to a brief conclusion paragraph, 1-2 sentences, as a main message of the paper, a summary of text currently written under conclusion.
--

REVIEWER	Ghulam Farooq Mansoor FHI 360, Afghanistan
REVIEW RETURNED	04-Nov-2017

GENERAL COMMENTS	The article has been substantially improved and could be accepted for publication. There are, however, some minor revisions suggested which may benefit quality of the manuscript. - The reviewer also provided a marked copy with additional comments. Please contact the publisher for full details.
--

VERSION 2 – AUTHOR RESPONSE

Responses to Reviewers

Page references are related to the marked document

Reviewer(s)' Comments to Author and Responses:

Reviewer: 1

Reviewer Name: Chandrashekhar T Sreeramareddy

Institution and Country: International Medical University, Malaysia

The revised MS adheres to the PRISMA and technical language confirms with the research paper.

A few minor issues

Methods: questionnaire was pretested, how was the questionnaire in six UN languages validated.

Although we do not use the word “validated” explicitly, we mention the following:

“The survey instrument [Web Annex 1] was developed by WHO with inputs from UNICEF and was tested in three different WHO regions to ensure questions were clear and appropriate and to estimate the time needed to complete the questionnaire.”

“Responses were tracked by WHO regional offices to maximize return and clarify inconsistencies or omissions. Filled in questionnaires were then returned from countries to WHO Headquarters.”

Pages 3 & 4, respectively

Need not use regional office (RO), may confuse with responsible officers.

We define RO on page 4 to avoid any confusion:

“These structured questionnaires were sent to a total of 130 WHO Member States by the six WHO Regional Offices (ROs) that implemented the survey through the WHO country offices (COs), provided support and follow-up.”

"As is common with self-administered questionnaires respondents may have inaccurately reported the level and extent of IMCI implementation, most likely reporting a more “positive” scenario than the reality." should be written under discussion, as limitations.

I agree. Because there was already a similar sentence in the conclusion, we have deleted it from the “methods session”

RESULTS: Eighty-nine per cent (82/94) when the number analysed was just 94, please have an alternative methods for presenting such number less than 100 where % do not make much sense.

We have rephrased it (page 5):

“Eighty-nine per cent (82 countries) of the 94 responding countries had an IMCI focal point at national level, whereas only 68% (62/91) had one at the sub-national (regional or district) level.

conclusion paragraph is way too long, most of which are implications or ways forward from this study findings. there needs to a brief conclusion paragraph, 1-2 sentences, as a main message of the paper, a summary of text currently written under conclusion.

We have now separated one first session for "Discussion" in which we now include some implications of the results obtained. It also includes some aspects of the findings such as coverage and scale up of IMCI, length of training courses, implementation of each component, etc. in light of the literature. And a second session with a paragraph for "Conclusion" that looks forward and links with important initiatives and efforts.

Pages 8-10

Reviewer: 2

Reviewer Name: Ghulam Farooq Mansoor

Institution and Country: FHI 360, Afghanistan

See file attached.

The article has been substantially improved and could be accepted for publication. There are, however, some minor revisions suggested which may benefit quality of the manuscript.

We thank the reviewer for the very useful comments and suggestions. They have been addressed in the text and suggested references have been added.

Reviewer: 3

Reviewer Name: Tanya Doherty

Institution and Country: South African Medical Research Council, South Africa

Competing Interests: I was a member of the study advisory group for the IMCI strategic review

Thank you for the opportunity to review a revision of this manuscript. The authors have addressed the reviewer comments well. There are just a few areas where I feel some minor changes are necessary:

- I do still believe that the methods section would benefit from some sub headings. Currently it is quite dense with a lot of information and long paragraphs. It would help readers to split this information into sub headings such as: study design, sampling, data collection, data analysis. Etc.

We have changed it.

Pages 3 & 4

- The statement at the end of the methods section “as is common with self-administered...” belongs in the limitations section of the discussion.

I agree. Because there was already a similar sentence in the conclusion, we have deleted it from the “methods section”

- The limitations section should come before the conclusion so that the conclusion provides the main messages and concluding thoughts only.

We have moved the paragraph on limitations up and have split the full session into two - “Discussion” and “Conclusion” to make it clearer

Pages 8 -10

- The authors present a new finding in the abstract and discussion: “These full implementer countries were 3.6 [95% CI 1.5 8.9] times more likely to achieve MDG4 than other (not full implementer) countries.” This result does not appear anywhere in the results section, only in the abstract and conclusion. Also the analysis methods used to generate this estimate are not included in the data analysis section. Either this result needs to be included in the results section and the analysis methods e.g. logistic regression or whatever method was used needs to be described in the data analysis section or it should be removed.

On page 5, under “Results”, we mention that “Forty-four countries have reported implementing IMCI in more than 90% of districts and also having all three IMCI components in place; these are considered full implementer countries. These countries are home to 160 million of the global under-five children.”

We have added one sentence with the OR results and corresponding 95% CI in this session

Page 6

- The issue of IMCI in emergency settings is raised in the conclusion but emergency settings are not mentioned in the results. Could the authors bring our results relevant to emergency settings in the results section so that readers can link the concluding recommendations with the study results.

We agree and have deleted it

- It is surprising that sustainability is not mentioned in the conclusions given the strong finding in figure two that high under-5 mortality countries had the least government contribution to the five elements of IMCI.

Yes, this is a good point. We have added a sentence addressing this issue.

Page 9

- The statement in the conclusion: “Our findings show that low income countries particularly hailed IMCI efficiency in programming and in service provision, as well as equity in access and coverage of intervention” is unclear. Please can this be re phrased.

We have deleted this sentence

- There are some typos and grammatical errors in the revised text which need a careful edit.

We have carefully revised the manuscript to reduce and hopefully eliminate typos and errors.

VERSION 3 – REVIEW

REVIEWER	Tanya Doherty South African Medical Research Council
REVIEW RETURNED	05-Dec-2017
GENERAL COMMENTS	The authors have addressed all previous comments and I have no further comments to make.